# Effect of Curing Temperature on Crack Resistance of Low-Heat Portland Cement Hydraulic Lining Concrete

**DOI:** 10.3390/ma18071618

**Published:** 2025-04-02

**Authors:** Shujun Chen, Xiangzhi Kong, Shuangxi Li, Bo Wei

**Affiliations:** 1College of Water Conservancy and Civil Engineering, Xinjiang Agricultural University, Urumqi 830052, China; csjfwz5201314@163.com (S.C.); xjlsx123@126.com (S.L.); 2China Institute of Water Resources and Hydropower Research, Beijing 100038, China; 3Xinjiang Shuifa Construction Group Co., Ltd., Urumqi 830000, China; 13899426936@163.com

**Keywords:** low-heat Portland cement concrete, mechanical property, maturity, curing temperature, microscopic test

## Abstract

As part of this study, mechanical property tests were carried out at different stages with different curing temperatures to elucidate the effect of temperature on the mechanical properties of concrete. The curing temperatures were laboratory curing temperature (standard curing at 20 °C) and variable temperature curing (simulated site ambient temperature curing) according to the actual temperature of previous construction sites. The compressive strength, split tensile strength, axial tensile strength, and modulus of elasticity values were tested, and the growth rates were calculated. According to previous experiments, the maturity indexes under two kinds of maintenance conditions were calculated based on the N-S maturity formula, F-P equivalent age calculation formula, and D-L equivalent age calculation formula proposed by the maturity theory. Moreover, logarithmic function, exponential function, and hyperbolic function fitting were carried out using the fitting software to study the developmental relationship between strength and maturity. The physical phase analysis of low-heat cement was performed using XRD and simultaneous thermal analysis, and pore structure analysis was conducted using the mercuric pressure method (MIP). We also conducted an SEM analysis of hydration products and the micromorphology of low-heat cement with 25% fly ash. Energetic spectroscopy analyzed the elemental content. In this study, it was found that temperature has a significant effect on the mechanical properties of concrete, with temperature having the greatest effect on splitting tensile strength. The strength of low-heat silicate cement concrete increases with maturity. The highest correlation coefficient was based on the hyperbolic function fit in the F-P equivalent age. The improved development of concrete strength in the later stages of the two curing conditions in this test indicates that low-heat cement is suitable for use in hydraulic tunnels. The low-heat cement generates a large number of C-S-H gels via C_2_S in the late stage, filling the internal pores, strengthening the concrete densification to make the structure more stable, guaranteeing the late development of concrete strength, and imparting a micro-expansive effect, which is effective for long-term crack resistance in hydraulic lining structures.

## 1. Introduction

The development of China’s water conservancy industry has led to advances in areas such as hydraulic lining structures. As a result, the study of water transfer tunnels has been continuously emphasized. This type of work often requires high durability and crack resistance in the structure itself because of the high volume and long distance involved in transportation. In order to improve the crack resistance of concrete, we must improve construction supervision and processes. More importantly, the improvement in concrete’s crack resistance begins with the raw materials involved. Low-heat Portland cement has low early strength and high late strength. Low-hydration heat and less dissipation of heat are some of the advantages of the slow rise in temperature occurring inside concrete [1]. This can effectively control the external and internal temperature difference between inside and outside of concrete, reduce temperature stress, and reduce the generation of cracks, and it has unique advantages in the context of crack resistance in concrete.

At present, many scholars have conducted research and performed demonstrations. Chen Gaixin [2] compared the performance tests of medium- and low-heat concrete, showing that low-heat cement can be safely used in dam structures. Panli [3] performed a coordinated control test of the temperature and deformation fields, and the study showed that low-heat cement can inhibit the temperature-induced cracking in mass concrete. L. Wang [4] investigated the mechanical properties, long-term hydration heat, drying shrinkage, and crack resistance of concrete in dam concrete with low-heat silicate mixed with fly ash. The concrete strength index is a key factor in the crack resistance of concrete structures. It is also the core technical parameter of engineering quality assurance. Li Xiangyu [5] showed that an increase in concrete tensile strength will increase the effects of axial tensile, splitting, and bending. Binbin Zhou [6] constructed a mechanical model for strength prediction based on the study of masonry cracks. Mustapha Jamaa Garba [7] carried out much mechanical property research in the context of the application of low-heat cement in dams. However, the research on hydraulic tunnels is slightly insufficient. Temperature is one of the factors that affect the mechanical properties of concrete. The relationship between curing temperature and strength has also been explored by many scholars. Guan Bin [8] studied the mechanism by which curing temperature influence the thermal conductivity of low-heat cement concrete. A prediction model of thermal conductivity considering the temperature effect is established. Shen Xin [9] studied the mechanism by which temperature influences low-heat cement by simulating a real environment. The low early strength of low-heat cement is also a concern in engineering applications. However, we can verify the very early-stage strength of low-heat concrete using the maturity method and ensure the applicability of the project. The strength development can also be predicted, and we can provide theoretical support for practical engineering applications. Nurse [10] and Saul [11] proposed a maturity function and equivalent age function. After many scholars’ research and improvements, Freiesleben Hansen [12] and Pedersen [13] proposed the equivalent age maturity calculation model based on the Arrhenius theory (hereinafter referred to as the F-P equivalent age formula). This formula is the most accurate and widely used. China has also studied this aspect for many years. Form DL/5144-2015 ”Hydraulic concrete construction specification” [14] provides the corresponding equivalent age calculation formula (hereinafter referred to as D-L equivalent age formula). Dai Jinpeng [15], through standard curing and negative temperature curing tests, calculated the maturity and fit the three fitting functions with the compressive strength. Zhong Yuehui [16] studied the relationship between early-age maturity and the strength of low-heat cement concrete. At present, the application of low-heat cement is mostly in its very early stages of maturity. However, there is insufficient research on the later stage of concrete.

Our test was conducted using standard curing for the first 30 d of aging and though the simulation of the actual temperature of the construction site. Then, 60 d isothermal curing was used. The temperature selected for this variable temperature curing is the actual temperature of concrete measured at the construction site. We explored the mechanical properties of low-heat Portland cement concrete at different stages under the same water/binder ratio, as well as the relationship between strength and maturity. The accuracy of the model is determined and analyzed using the fitting function. The application of low-heat cement in the hydraulic tunnel provides a theoretical basis. The low-heat cement mixed with 25% fly ash was studied using a microscopic test. Further, we explained the change law of the mechanical properties.

## 2. Experiment

### 2.1. Raw Materials and Mix Proportion

#### 2.1.1. Raw Material

The raw material used in this test was 42.5 low-heat Portland cement (hereinafter referred to as low-heat cement), referred to as LHC (namely, ASTM C150 (2018)-type IV Portland [17]). Low-heat cement is a water-hard cementitious material with a low hydration heat made from silicate cement clinker of an appropriate composition, together with an appropriate amount of gypsum and finely ground cement. We used Class F–Grade I fly ash, secondary natural sand, small and medium stones, a standard superplasticizer, and an air-entraining agent. The solid content of the water reducer was 15.48%. The solid content of the air-entraining agent was 58.5%. Low-heat cement has a slow setting time and low hydration heat. The detection of cement and fly ash is shown in Table 1 and Table 2 below. The coarse aggregate was broken pebbles, and the fine aggregate was natural sand. The aggregate test results are shown in Table 3. The raw materials used in this experiment were selected according to the previous construction materials and reading the literature.

#### 2.1.2. Mix Proportion

This test was based on the absolute volume method in SL/T352-2020 “Hydraulic Concrete Test Procedures” [18]. We mixed C_90_35 secondary pumping concrete in an indoor. The slump was at maintained 180~220 mm. The gas content was maintained at 4.0%~5.0%. Fluidity and cohesion improved. The proportions of low-heat silicate cement hydraulic lining concrete No. DRH are shown in Table 4.

### 2.2. Test Scheme

#### 2.2.1. Variable Temperature Curing Test

This test, conducted at the standard maintenance temperature (20 °C), consisted of a box of the proposed environmental temperature to simulate the actual temperature of the site for variable temperature maintenance. The standard curing specimen number is DRH-I. The specimen number of quasi-environmental maintenance is DRH-II. The first 30 days of aging were spent in curing at different temperatures. The same standard maintenance was carried out in the following 60 days. The curing temperature is shown in Figure 1.

#### 2.2.2. Mechanical Performance Test

DRH (1 d, 3 d, 7 d, 14 d, 28 d, 56 d, 90 d) was used for compressive, modulus of elasticity, split tensile, and axial tensile strength tests. A YAW-3000 microcomputer control electro-hydraulic servo pressure testing machine (Dawson Group Ltd., Qingdao, China) was used on a 150 × 150 × 150 mm concrete cube test block for compressive strength testing. We made a Φ100 × 300 mm concrete cylinder specimen, and the test machine was the same as that used in the concrete compression test. We performed modulus of elasticity tests. We made a 150 × 150 × 150 mm concrete cube test block, then used a WAW-600 microcomputer-controlled electro-hydraulic servo universal testing machine for splitting tensile strength testing. We used a WDW-100-type microcomputer-controlled electronic testing machine on 550 × 130 × 100 mm concrete eight-mold test blocks for axial tensile strength testing. The mechanical property tests conducted at each stage were averaged from three results of the test indexes. Compressive strength and split tensile strength require that the median value is taken as the result when there is a measured value and median value difference of more than 15%, and when the difference between two measured values and the median value is more than 15% of the median value, the data are invalid. Testing the axial tensile and modulus of elasticity strengths required the average be taken when one of the measured values exceeded 15% of the average value, and when the difference between the two measured values and the average value exceeded 15% of the average value, the data are invalid. The tests conducted and the specimens produced are from the “Hydraulic Concrete Testing Regulations” (SL/T 352-2020).

#### 2.2.3. Maturity Calculation

Through the N-S maturity calculation formula, F-P equivalent age calculation formula, and D-L equivalent age calculation formula, three maturity calculation functions were obtained. Three fitting functions, including the exponential logarithmic and hyperbolic functions, were fitted. We determined the most accurate fitting relationship between maturity and strength, and we verified the relationship between strength and maturity. We verified that maturity is applicable to hydraulic lining concrete and that early-stage strength predictions can be made for concrete.

#### 2.2.4. Microscopic Tests

The physical phase analysis of low-heat cement at different curing stages used an X’Pert PRO-type X-ray diffractometer (XRD). We scanned the 2*θ* angle in the range of 10–80°at a scanning speed of 10°/min, with an operating voltage of 40 kV. The crystalline phase data were processed, and XRD was plotted using MDI jade6 software. Simultaneously, we used a thermal analyzer using NETZSCH 5 (Guangzhou, China), under a nitrogen atmosphere, with heating temperatures from 50 to 1000 °C and heating rates of 10 °C/min. We performed thermogravimetric (TG) and differential scanning calorimetry (DSC) analyses. We also performed pore structure analysis using a MicroActive AutoPore V 9600 Mercury-in-Pressure (MIP) meter (Micromeritics, Norcross, GA, USA). We tested the microstructural characteristics and hydration product distribution patterns of low-heat cement at different curing stages using a JSM-7900F scanning electron microscope (SEM) (JEOL, Tokyo, Japan). The atomic content of the sample crystals was further analyzed using an X-ray photoelectron spectrometer (EDS) of the XPS-ESCALAB type. XRD, MIP, SEM, and EDS were used for 7 d, 28 d, and 90 d age detection. Synchronized thermal analysis was used for 90 d age detection

## 3. Experiment Results and Analysis

### 3.1. Mechanical Properties Analysis

#### 3.1.1. Compressive Strength

Two maintenance methods for the compressive strength of concrete meet the design requirements. Figure 2 shows the compressive strength of concrete with two curing methods on the left side. The simulated temperature is lower than the standard temperature before 1 day at the curing temperature. Therefore, DRH-II is 1.7 MPa lower than DRH-I as the temperature of the quasi-environmental maintenance box rises. At the ages of 3 d and 7 d, DRH-II was 1.3 MPa and 2.1 MPa higher than DRH-I, respectively. With the decrease in temperature, the intensity value of DRH-II was lower than that of DRH-I. Later, when cured at the same temperature, the difference between the two strength values was small. The right side of Figure 2 is the growth rate of the compressive strength. Based on the 28 d age, DRH-II was 3.6% lower than DRH-I at the 1 d age. At the 3 d and 7 d ages, DRH-II was 5.7% and 7.1% higher than DRH-I, respectively. The growth rate is roughly the same when curing at the same temperature in the later period. At 90 days of age, DRH-II was 11% lower than DRH-I. On the whole, the strength development and strength growth rate change with the increase and decrease in temperature. It can also be concluded that the early strength growth of DRH is better than that in the context of laboratory-standard curing. This is beneficial to cracking the resistance of concrete at an early stage. Increased compressive strength also means increased concrete compactness and resistance to shrinkage.

#### 3.1.2. Elastic Modulus

Elastic modulus is also an important index in the mechanical properties of concrete. This test is maintained by two curing temperature methods. The corresponding elastic modulus of concrete and its growth rate are obtained, as shown in. Figure 3. The left side of the figure shows the elastic modulus of concrete, which is basically the same in terms of development as compressive strength. Among them, at 3 d and 7 d of age, DRH-II was 2.5 GPa and 1 GPa higher than DRH-I, respectively. The right side of the figure is the growth rate of the elastic modulus of concrete. Taking 28 d age as the reference value, at 3 d and 7 d of age, the growth rate of DRH-II was 11.4% and 6.6% higher than that of DRH-I, respectively. As the temperature decreases, the growth rate also decreases. The growth rate of DRH-II is still lower than that of DRH-I when cured at the same temperature. At the age of 90 days, the difference between the two was 2.3%. As with the compressive strength, it is found that the temperature change has an effect on the compressive index of concrete mechanical properties. The continuous and stable growth of the modulus of elasticity can resist the deformation of concrete due to drying shrinkage and so on. This is the same as the Zhou Shihua [19] study.

#### 3.1.3. Splitting Tension

Splitting tensile strength is also one of the crack resistance indexes of concrete. The left side of Figure 4 shows the splitting tensile data value. The simulated temperature is high at 3 d and 7 d of age. The strength of DRH-II is 0.3 MPa and 0.3 MPa higher than that of DRH-I. The overall strength increases and decreases with the rise and fall in temperature. The right side of Figure 4 shows the growth rate of the splitting tensile strength. Taking 28 d of age as the reference value, the early growth rate is basically the same as the compressive growth rate. The growth rate of DRH-II was 11.4%, 12%, and 3.5% higher than that of DRH-I at 3 d, 7 d, and 14 d. Isothermal maintenance was at a later stage, and the DRH-II growth rate was almost identical to DRH-I development. The variable-temperature curing, in this case, is simulated based on the temperature of the construction site, which indicates that the indexes of split tensile strength derived in the laboratory are more in line with reality. This reflects the fact that split tensile strength is more sensitive to temperature changes.

#### 3.1.4. Axial Tensile Strength

The axial tensile strength of concrete is very important to the crack resistance of concrete. The left side of Figure 5 shows the axial tensile strength of concrete. The strength of DRH-II was 0.1 MPa and 0.21 MPa higher than that of DRH-I at 3 d and 7 d of age. The right side of the figure shows the growth rate of the axial tensile strength of concrete. The age of 28 d is the reference value. The growth rate of DRH-II was 2.8% and 9.7% higher than that of DRH-I at 3 d and 7 d. In the later period, the growth rate decreased due to the decrease in temperature. However, at the age of 90 d, the growth rate of DRH-II was 1.3% higher than that of DRH-I. Concrete has the characteristics of compressive strength and tensile strength. Combined with splitting tensile strength, it shows that the temperature change is more sensitive to the tensile strength of concrete.

### 3.2. Maturity Theoretical Analysis

Combined with the above, we can conclude that the increase in curing temperature will increase the strength of concrete. To further analyze the relationship between curing temperature and strength, according to previous experience and relevant norms, we then used three maturity calculation formulas to calculate the maturity index. The exponential, logarithmic, and hyperbolic functions serve as the three types of function fitting. The highest precision is selected to represent the relationship between maturity and strength development.

#### 3.2.1. Maturity Indicator

Maturity means that the strength of concrete is a function of the product of curing age and temperature, and its strength is approximately the same when the product of different ages and temperatures are equal; the method of calculating the strength of concrete using this product becomes the maturity method.

N-S maturity calculation formula [20]:(1)$$M=\sum_{0}^{t}(T-T_{0})\Delta t$$

M—maturity, °C·d; T—actual temperature of concrete, °C; T_0_—reference temperature (−10 °C); Δt—curing age, d.

P equivalent age [21]:(2)$$T_{e}=\sum_{0}^{t}exp[\frac{E}{R}(\frac{1}{273+T_{c}}-\frac{1}{273+T})]\Delta t$$

T_e_—equivalent age, d; E—activation energy, J/mol (T ≥ 20 °C, E = 33,500 J/mol; T < 20 °C, E = 33,500 + 1470 (20 − T) J/mol)); R—gas constant (8.3144 J/(mol·K)); T_c_—reference temperature (20 °C); T—actual temperature of concrete, °C; Δt—time interval, d.

L equivalent age [22]:(3)$$t=\sum \alpha_{T}t_{T}$$

T—equivalent age, d; $\alpha_{T}$—the temperature is the equivalent coefficient of T; $t_{T}$—duration of temperature T, h.

We check whether the specification can be obtained at 20 °C. The D-L equivalent coefficient is 1.0. The rest also must also be known, (1)–(3). The maturity index at different curing temperatures can be calculated. The calculation results of the maturity index are as follows in Table 5. The F-P equivalent age and D-L equivalent age are the same in standard curing. The curing is according to on-site temperature. The equivalent age calculated by the F-P equivalent age calculation model and the D-L equivalent age calculation model are relatively close.

#### 3.2.2. The Relationship Between Maturity and Strength

After the maturity value is obtained, the maturity strength relationship fitting equation can be determined. The exponential, logarithmic, and hyperbolic curves were used to fit the compressive strength, splitting tensile strength, axial tensile strength, and maturity, respectively.

Using the exponential function, according to Freiesleben and Pedersen [23], it is found that there is a relationship between hydration heat and maturity as well as between maturity and strength:(4)$$S=S_{\infty }e^{{-(\frac{\tau }{M})}^{a}}$$

S—concrete strength, MPa; M—maturity, (°C·h) or h; $S_{\infty }$—final strength of concrete, MPa; τ—temporal characteristic parameters, (°C·h) or h; a—Shape factor.

Using the logarithmic function, Plowman [24] proposed the following based on the linear relationship between strength and maturity:(5)S=a+blog(M)

S—concrete strength, MPa; M—maturity, (°C·h) or h; a, b are obtained by function fitting.

Using the hyperbolic function, Kee [25] found that the relationship between maturity and strength can also be expressed by a hyperbola:(6)$$S=\frac{M}{mM+n}$$

S—concrete strength, MPa; M—maturity, (°C·h) or h; m, n are obtained by function fitting.

#### 3.2.3. The Establishment and Analysis of Function Model

Using fitting software (4)–(6), the relationship curve between the compressive strength and maturity of low-heat Portland cement concrete can be obtained, as shown in Figure 6. The relationship curve between splitting tensile strength and maturity is shown in Figure 7, the relationship curve between axial tensile strength and maturity is shown in Figure 8, and the relationship curve between elastic modulus and maturity is shown in Figure 9. Overall, the compressive strength, splitting tensile strength, axial tensile strength, and elastic modulus of DRH-I and DRH-II increase with the increase in maturity index. The fitting curves obtained by the three fitting methods of F-P equivalent age and D-L equivalent age are roughly consistent. It shows that both can accurately predict the development law of concrete strength.

The hyperbolic function and logarithmic function in the fitting relationship between maturity and strength development can be well fitted. The effect is not good. The correlation coefficients are shown in Table 6. The fitting effect of the hyperbola is the best, and the logarithmic function fitting followed. Therefore, the hyperbolic function fitting model is the most accurate. It can be seen from the relationship between the four indexes of concrete mechanical properties and maturity that the fitting relationship between splitting tensile strength and maturity is the best. Alongside the results of the variable temperature curing test, the following conclusions can be drawn. The splitting tensile strength is most affected by temperature change. The correlation coefficients between the hyperbolic function and the split tensile strength were 0.96, 0.99, and 0.98. The fit is more accurate at early stages. Therefore, in practice, the influence of splitting tensile strength on concrete should be of greater concern. It can also be seen that the hyperbolic function fitting based on F-P equivalent age can be used for low-heat silicate cement hydraulic lining concrete for early-stage strength prediction.

### 3.3. Microscopic Analysis

We performed analyses of the micro-morphology, hydration products, and later-stage Ca(OH)_2_ content of low-heat cement blended with 25% fly ash material system (No. 2D). The test samples used in this test were 3 × 3 × 3 cm cubes based on microscopic ratios. These were soaked and maintained after solidification. Upon the completion of the aging process, the outer epidermis of the sample was peeled off, and the samples were soaked in anhydrous ethanol to abort hydration. Samples for XRD and simultaneous thermal analysis need to be milled to a powder of less than 80 µm. The samples used for MIP, SEM, and EDS need to be crushed into 1 × 1 × 1 cm pieces. Drying was performed in a vacuum-drying oven, and a final seal was performed prior to inspection. The microscopic test matches are summarized in Table 7.

#### 3.3.1. X-Ray Diffraction Analysis

We conducted an exploration of Ca(OH)_2_ in the hydration products and C_2_S in low-heat cement therein. Moreover, processing of crystalline phase data with Materials Date, Inc. JADE software followed, as shown in Figure 10. Then, we conducted XRD plotting. As shown in Figure 10, the XRD pattern of low-heat cement with 7 d, 28 d, and 90 d aging was numbered 2D7d, 2D28d, and 2D90d, respectively. The C_2_S diffraction peaks in Figure 10 found near 25° and 45° were detected, and Ca(OH)_2_ diffraction peaks were found near 20° and 32°; Ca(OH)_2_ and C_2_S crystal strengths were found to be more prominent near 20° and 30°. The Ca(OH)_2_ diffraction peak at 90 d was slightly decreased compared to that at 28 d. It is hypothesized that perhaps the low-heat cement produces more of a hydrated calcium silicate (C-S-H) gel than Ca(OH)_2_ at a later stage. The advantages of the high C_2_S content and low C_3_S content of low-heat cement involve reduced amounts of the hydration product Ca(OH)_2_ and more C-S-H gel, with the hydration equations shown in Equation (7) [26]. C_2_S was detected at all stages. This suggests that low-heat cement can be sufficiently hydrated to produce C-S-H gels at a later stage. In the cement, this fills pores so that the low-heat concrete structure is denser, which causes the later-obtained mechanical properties to continually improve, and this is conducive to the long-term crack resistance of the lining structure.(7)$$2C_{2}S+9H\to C_{3}S_{2}H_{8}+CH$$

#### 3.3.2. Simultaneous Thermal Analysis

The test results are shown in Figure 11. Figure 11a shows the TG-DTG plot. It can be seen that the quality of the low-heat cement decreases as the temperature increases. Sharp weight loss occurred at 400 °C. From the DTG, it can be seen that the rate of decline in low-heat cement is relatively flat. This indicates a more stable structure in the low-heat cement system. Figure 11b shows the TG-DSC plot; heat loss by heat absorption occurs at 400 °C in the heat flow curve, suggesting that 90 d old Ca(OH)_2_ decomposes due to heat absorption at this time and that Ca(OH)_2_ content decreases in later stages. DSC’s heat absorption peak matches the TG weight loss peak. The onset and termination temperatures of the decomposition of Ca(OH)_2_ crystals can be calculated to give the Ca(OH)_2_ content per 100 g of the sample. The Ca(OH)_2_ of the low-heat cement paste sample was 8.23 g.

#### 3.3.3. Pore Structure Analysis

The test results are shown in Figure 12. Figure 12a shows the pore size distribution of low-heat cement at 7 d, 28 d, and 90 d, numbered 2D7d, 2D28d, and 2D90d. The pore size of low-heat cement is concentrated around 100–110 nm at 7 d of age, mainly around 50 nm at 28 d of age, and 10–20 nm at 90 d of age. As shown in Figure 12b, 7 d, 28 d, and 90 d porosity distributions of low-heat cement are numbered 2D7d, 2D28d, and 2D90d, respectively. The early hydration reaction is slow, and the pore size and porosity in low-heat cement are larger, so the early mechanical properties are weaker. However, as age progresses, the hydration reaction of C_2_S in low-heat cement at 90 d produces Ca(OH)_2_ and C-S-H gels; a large number of pore spaces are filled, making the concrete structure dense, so the mechanical properties of concrete are greatly improved later. With the law of evolution of porosity and pore size of low-heat cement, the porosity and pore size of low-heat cement will decrease even more in the later stage, and the DRH strength will continue to increase. According to XRD and simultaneous thermal analysis, it can be seen that the Ca(OH)_2_ content is less at 90 d. It is therefore hypothesized that low-heat cement produces more C-S-H gels at a later stage.

#### 3.3.4. Scanning Electron Microscope

The results are shown in Figure 13, Figure 14 and Figure 15. It is found that Ca(OH)_2_ in low-heat cement is mostly stacked in flakes. There is good compactness throughout the early to later stages. It is surrounded by C-S-H gel. The early C-S-H gel of low-heat cement is distributed sporadically in the form of agglomerates, so the early mechanical properties of DRH are low. Later on, by virtue of the high content of C_2_S, it gradually wraps the fly ash and Ca(OH)_2_ tightly in a fine mesh. The ability to fill pores can make the concrete structure more stable. It also retains a considerable amount of Ca(OH)_2_ to ensure late strength development. This is beneficial to the continuous improvement of the mechanical properties; maintaining the overall cracking resistance to improve the long-term cracking resistance of concrete is important. Some calcite was found at 90 d, but it was mostly small and scattered and would not undergo expansion damage.

#### 3.3.5. Energy Spectrum Analysis

We tested the low-heat cement plus 25% fly ash cementitious systems at different curing stages. Energetic spectral elements were detected, as shown in Table 8. For a more detailed understanding of the elemental changes in hydration product crystals, we tested the 7 d, 28 d, and 90 d conservation-aged samples using energy spectrum analysis. Seven fixed-point tests were performed on cement samples at each age, and the average value was taken. The 7 d–90 d low-heat cement samples were numbered D1, D2, D3. Low-heat cement also retains a certain degree of Ca and Si elements in later stages and is able to produce C-S-H gels during long-term hydration. This causes a denser concrete structure that is conducive to later strength. In addition to the findings in other microscopic tests, we found a slight decrease in Ca(OH)_2_ peak at 90 d for low-heat cement. However, the strength increases in the later stages; thus, in these stages, the low-heat cement generates more C-S-H gels. The low content of Al and S elements, in combination with SEM, suggests that less AFt was generated in the later stage. A certain amount of Mg is still present at 90 d. Hydration generation produces Mg(OH)_2_ and a micro-expansive effect in concrete, inhibits self-shrinkage, and improves the durability of concrete.

## 4. Conclusions

In this experiment, it was found that temperature has a significant effect on the early mechanical properties of concrete, more specifically, mechanical properties and their growth rates in terms of standard and proposed environmental curing. Mechanical properties increase with increasing temperature and decrease with decreasing temperature. Temperature changes have the greatest effect on the tensile aspect of concrete in terms of splitting resistance.In this experiment, it was found that the late mechanical properties of low-heat silicate hydraulic lining concrete under the two curing methods are considerable and can be applied to hydraulic tunnels.Based on the highest correlation coefficient of the double curve fitting function in the F-P equivalent age and D-L equivalent age, the maturity can be applied to the early-age-strength prediction of hydraulic lining concrete with low-heat silicate cement, which provides a reference scheme for the demolding of hydraulic tunnels.Microscopic tests show that low-heat cement generates a large number of C-S-H gels at a later stage, which makes the concrete structure dense and has a micro-expansive effect. Thus, an improvement in strength and crack resistance is imparted over time.

## Figures and Tables

**Figure 1 materials-18-01618-f001:**
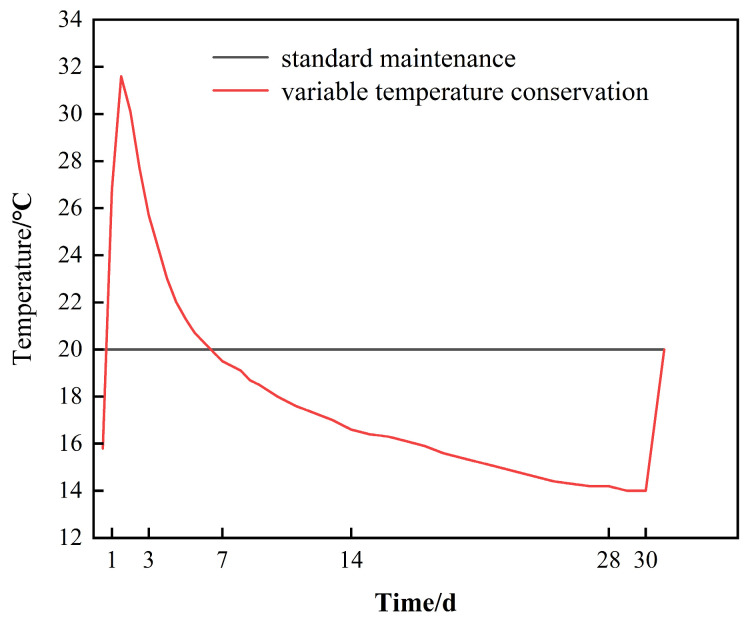
Curing temperature diagram.

**Figure 2 materials-18-01618-f002:**
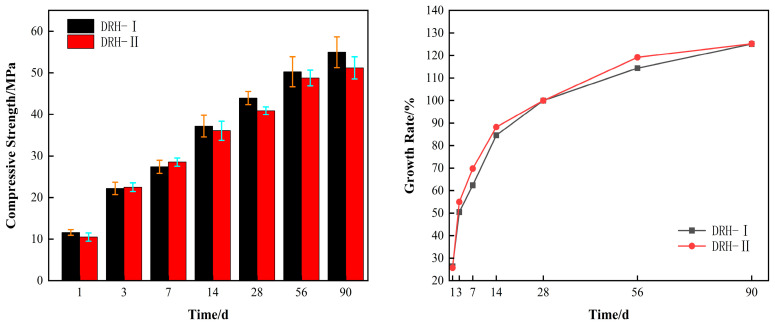
Compressive strength and growth rate.

**Figure 3 materials-18-01618-f003:**
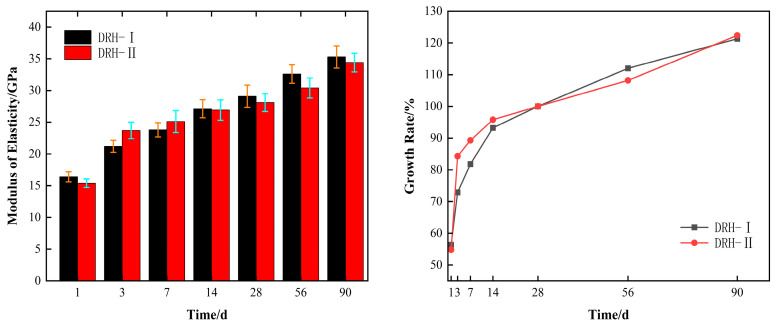
Modulus of elasticity and growth rate.

**Figure 4 materials-18-01618-f004:**
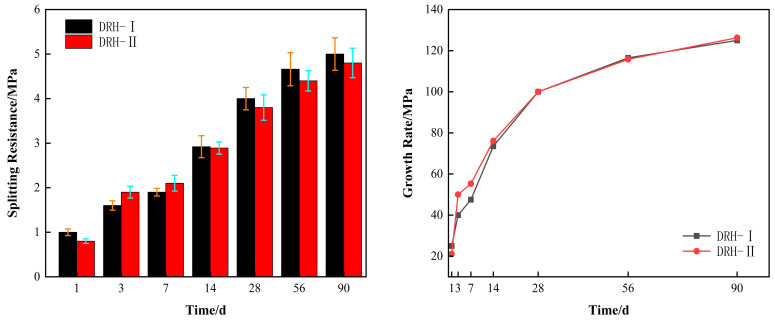
Splitting tensile strength and growth rate.

**Figure 5 materials-18-01618-f005:**
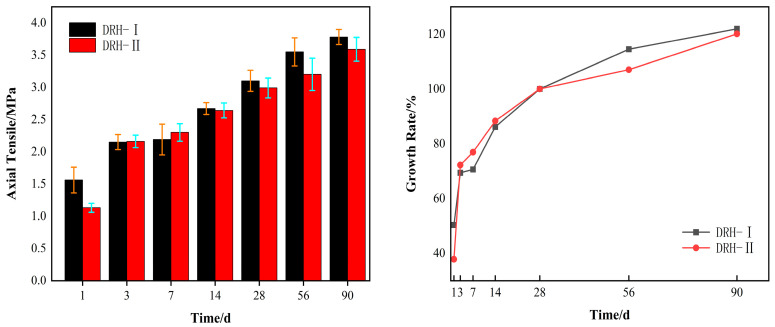
Axial tensile and growth rate.

**Figure 6 materials-18-01618-f006:**
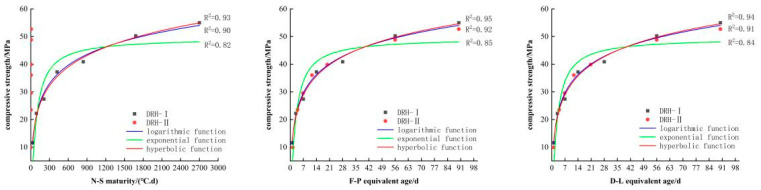
Relationship between compressive strength and maturity index.

**Figure 7 materials-18-01618-f007:**
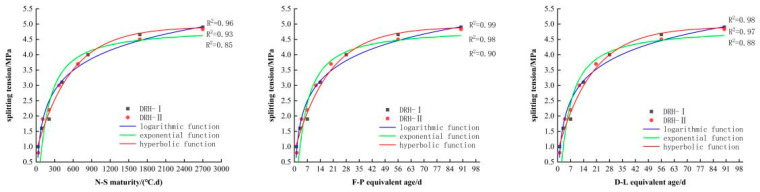
Relationship between splitting tensile strength and maturity.

**Figure 8 materials-18-01618-f008:**
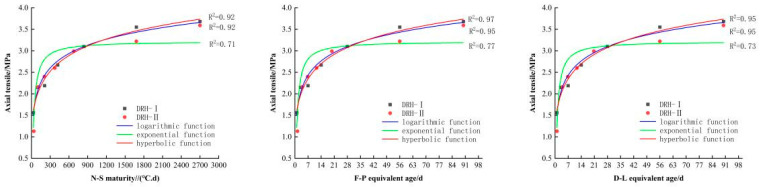
Relationship between axial tensile strength and maturity index.

**Figure 9 materials-18-01618-f009:**
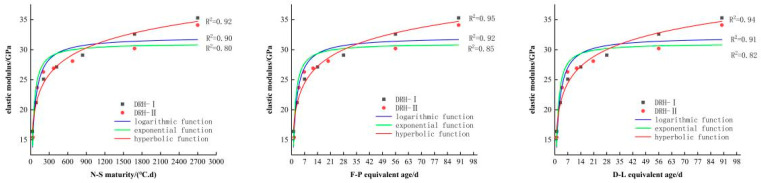
Relationship between elastic modulus and maturity index.

**Figure 10 materials-18-01618-f010:**
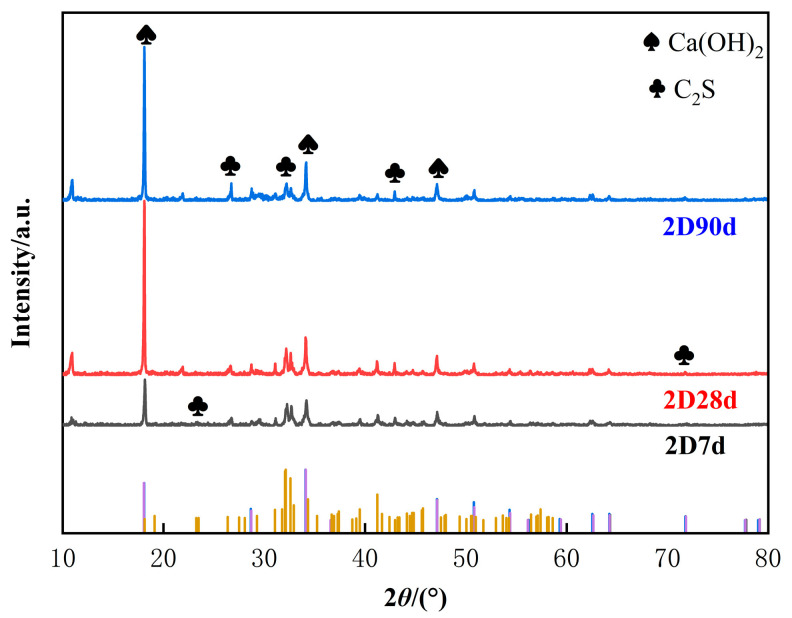
XRD pattern.

**Figure 11 materials-18-01618-f011:**
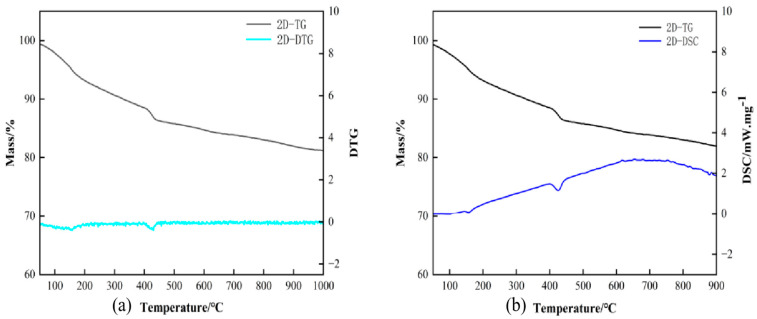
TG-DTG and TG-DSC diagrams for low-heat cements.

**Figure 12 materials-18-01618-f012:**
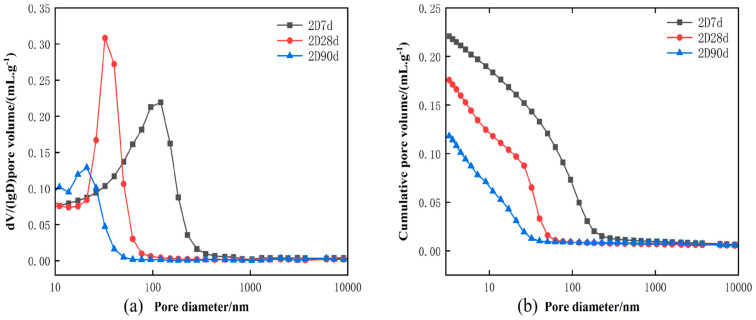
Low-heat cement pore size vs. porosity plot.

**Figure 13 materials-18-01618-f013:**
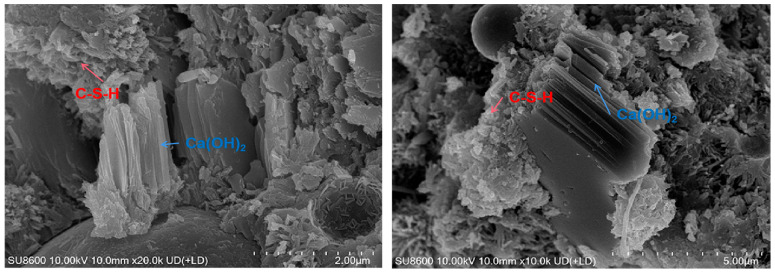
Hydration products of low-heat cement with 25% fly ash system (7 d).

**Figure 14 materials-18-01618-f014:**
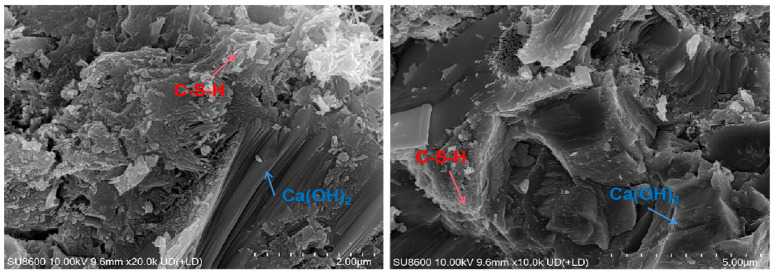
Hydration products of low-heat cement with 25% fly ash system (28 d).

**Figure 15 materials-18-01618-f015:**
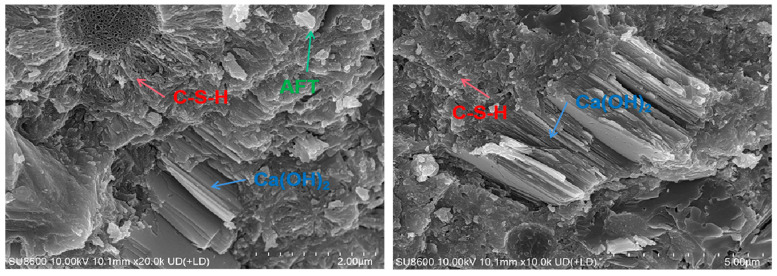
Hydration products of low-heat cement with 25% fly ash system (90 d).

**Table 1 materials-18-01618-t001:** Chemical composition detection of cement and fly ash, %.

Specimen	CaO	SiO_2_	Al_2_O_3_	MgO	Fe_2_O_3_	SO_3_	F-CaO	Loss	K_2_O	Na_2_O	R_2_O
LHC	59.8	23.04	4.42	3.2	4.30	2.52	0.84	1.09	0.56	0.26	0.63
Fly ash	13.18	50.05	16.1	4.5	5.5	1.11	0.71	2.16	3.64	1.4	4.56

**Table 2 materials-18-01618-t002:** Testing of physical properties of cement.

Cement	Specific Surface Area/m^2^·kg^−1^	Density/g·cm^−3^	Normal Consistency/%	Setting Time/min		Hydration Heat/KJ·kg^−1^		Compressive Strength/MPa				Break off Strength/MPa			
				Initial Set	Final Set	3 d	7 d	3 d	7 d	28 d	90 d	3 d	7 d	28 d	90 d
LHC	317.6	3.23	27.7	216	291	188	220	17.4	30.3	51.3	67.4	3.9	5.0	7.0	8.2

**Table 3 materials-18-01618-t003:** Aggregate performance testing.

Aggregate	Coarse Aggregate		Fine Aggregate
Grain Size/mm	20–40	5–20	≤5
Dry apparent density of saturated surface/kg/m^3^	2680	2650	2630
Water absorption at saturated surface–dry basis/kg/m^3^	0.42	0.63	0.80
Soil content/%	0.5	0.2	1.4
Fineness modulus	-	-	2.8

**Table 4 materials-18-01618-t004:** Proportioning of concrete.

Numbering	Water-Binder Ratio	Fly Ash/%	Admixture/%		Amount/kg·m^−3^					
			Water Reducing Admixture	Air Entraining Agent	Water	Cement	Fly Ash	Sand	Pebble	Nakaishi
DRH	0.36	25	1	0.006	140	291.7	97.2	771.6	566.8	469

**Table 5 materials-18-01618-t005:** Maturity index calculation results.

	DRH-I							DRH-II						
Time/d	1	3	7	14	28	56	90	1	3	7	14	28	56	90
Temperature/°C	20							26.8	25.7	19.5	16.6	14.2	20	20
N-S/(°C·d)	30	90	210	420	840	1680	2700	31.3	108.9	233.7	428	780.4	1608	2628
F-P/d	1	3	7	14	28	56	90	1.078	4.074	8.314	14.54	25.134	52.5	86.5
D-L/d	1	3	7	14	28	56	90	1.089	4.097	8.356	14.62	25.5	53	87

**Table 6 materials-18-01618-t006:** Correlation coefficient table.

Mechanical Property	Compressive Strength			Split Tensile Strength			Axial Tensile Strength			Elastic Modulus		
Maturity Method	N-S	F-P	D-L	N-S	F-P	D-L	N-S	F-P	D-L	N-S	F-P	D-L
logarithmic function	0.9	0.92	0.91	0.93	0.98	0.97	0.92	0.95	0.95	0.90	0.92	0.91
exponential function	0.82	0.85	0.84	0.85	0.90	0.88	0.71	0.77	0.73	0.80	0.85	0.82
hyperbolic function	0.93	0.95	0.94	0.96	0.99	0.98	0.92	0.97	0.95	0.92	0.95	0.94

**Table 7 materials-18-01618-t007:** Microtest ratios.

Number	Water-to-Cement Ratio	Fly Ash/%	Amount of Adhesive Material/g		
			Water	Cement	Fly Ash
2D	0.36	25	180	375	125

**Table 8 materials-18-01618-t008:** Energetic element test results.

Measurement Point	Time/d	Elemental Mass Percentage/%									Ca/Si
		C	O	Mg	Al	Si	S	Ca	Fe	other	
D1	7	9.45	37.35	0.36	1.07	11.34	0.83	37.95	1.42	0.23	2.35
D2	28	12.35	41.07	0.85	1.4	11.3	1.05	29.45	2.02	0.51	1.83
D3	90	11.23	41.3	0.52	2.73	12.19	1.12	29.14	1.77	-	1.67

## Data Availability

The original contributions presented in the study are included in the article, further inquiries can be directed to the corresponding author.

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
