# Peer review of "Effect of Curing Temperature on Crack Resistance of Low-Heat Portland Cement Hydraulic Lining Concrete"

_materials, 2025, doi:10.3390/ma18071618_

Round 1
Reviewer 1 Report (Previous Reviewer 1)
Comments and Suggestions for Authors
Dear authors, this is the third version of the article I review. However, you diligently ignore my recomendations.
What are the exact settings of XRD diffractometer (type and model of the detector, exposition time, voltage and current)? "Jade software" requires a reference.
Authors provided XRD patterns shown at figure 10 (left side). Right side of the figure remains unexplained. It totaly differs from the classic XRD patterns to the left. If these are calculated data extracted from some database they shouldn't differ from the age of the sample. The second issue is the broadness of the lines. It seems that the FWHM (or W) value of calculated pattern is too big. If one will set it to about 0.1 we'll see standard XRD patterns of Ca(OH)2 and C2S.
What is the meaning of rainbow colours at the right figure 10?
Comments on the Quality of English Language
Due to the poor level of English the article is almost unreadeble.
Here are just a few examples out of many:
line 19. Calculate growth rate. According to previous experience.
line 25. Energy spectrum analysis of element content.
line 44. Long haul distance, Therefore, the structure itself has high requirements for durability and crack resistance.
line 48. Low hydration heat, Less heat release, The advantages of slow temperature rise inside concrete.
Verbs are missing. The sentences are strangely separated by points.
The entire article requires careful proofreading. This is neither thereviewer's nor the editor's job.
Author Response
Please see the attachment.

Reviewer 2 Report (Previous Reviewer 2)
Comments and Suggestions for Authors
Dear Authors,
After reviewing the revised version of your manuscript, I can confirm that the comments have been adequately addressed. The improvements made have significantly enhanced the clarity and scientific rigor of the document.
Author Response
Please see the attachment.

Reviewer 3 Report (New Reviewer)
Comments and Suggestions for Authors
Dear Authors,
Thank you for your submission. Your study on the effects of curing temperature on the crack resistance and mechanical properties of low-heat Portland cement concrete provides valuable insights, particularly regarding the relationship between maturity and strength development. However, several aspects of the manuscript need significant improvement before it can be considered for publication.
Major Concerns:
- Lack of Portland Cement Specification: The type of Portland cement used (CEM I or CEM II) is not specified (line 99). This is critical as different cement types have different hydration characteristics and heat evolution properties.
- Curing Conditions – Missing Relative Humidity Information: While the temperature conditions are provided, there is no mention of relative humidity during curing. This is a crucial parameter that significantly influences hydration and mechanical properties.
- Missing Standard Deviations in Results: The manuscript lacks standard deviations for mechanical property measurements (compressive strength, tensile strength, elastic modulus, etc.). Including these values is essential for assessing the variability and reliability of the results.
- Axial Tensile Strength – Unusual Specimen Dimensions: The test specimens for axial tensile strength are reported as 600 × 196 × 100 mm, which seems non-standard. Please clarify whether this is a standard test size or if it was chosen based on previous studies.
- Microscopic Analysis – Missing Sample Preparation Details: The preparation of samples for SEM and EDS analysis is not described. This is critical, as improper preparation (e.g., unpolished surfaces) could lead to inaccurate results, particularly in EDS quantitative analysis.
- EDS Analysis – Missing Experimental Parameters: The manuscript does not provide details on the acceleration voltage, current settings, or acquisition time for EDS measurements. These parameters are crucial for the reproducibility and accuracy of the elemental analysis.
- SEM Figures – Missing Annotations: The SEM images should clearly indicate what is being shown (e.g., C-S-H gel, portlandite, unhydrated cement particles, pores). This would improve the interpretability of the images.
- EDS Data – No Direct Spectrum Comparison: The EDS spectra should be presented in a single graph for comparison instead of separate figures. This would provide a clearer view of compositional differences over time.
- Discussion Section – Weak Comparisons with Literature: While the results are presented, there is a lack of discussion comparing the findings to previous research. A critical analysis of how your data aligns or differs from existing studies would strengthen the manuscript.
Minor Suggestions:
- Clarify the significance of the hyperbolic function as the best fitting model for the maturity-strength relationship. Why does it perform better than the exponential or logarithmic models?
- Ensure figure captions provide enough context so they can be understood without referring to the main text.
- Explain the choice of curing temperatures in more detail—how do they compare to real-world conditions?
The manuscript contains multiple typographical and grammatical errors, including missing spaces, incorrect punctuation, and inconsistent formatting. Additionally, some sentences are unclear or awkwardly structured, making the text difficult to follow in certain sections.
Key areas for improvement:
- Typographical errors: Missing spaces, incorrect punctuation, and inconsistent formatting (e.g., “Flyash” instead of “Fly ash” in Table 4).
- Acronyms should be defined at first use: For example, "EDS" appears in Section 2.3 without prior explanation. Ensure that all abbreviations (SEM, EDS, XRD, etc.) are introduced properly when first mentioned.
- Sentence structure: Some sentences are too long or poorly structured, which affects readability. Consider revising them for clarity and conciseness.
- Technical terminology: Certain terms could be used more precisely, particularly in the methodology and results sections.
A thorough proofreading by a native or proficient English speaker is strongly recommended to improve clarity and coherence.
Round 2
Reviewer 1 Report (Previous Reviewer 1)
Comments and Suggestions for Authors
Dear authors!
You've definitely made a serious work to improve the article.
I've found a few mistakes, that shoud be corrected.
line 142 "We make a" Perhaps, it should be "We made "
line 169 "We scanning" should be ""We scanned"
line 208 "the same in tetms" should be "the same in terms"
line 254 "hyperbolic functions sreve as the three types" I have no idea what sreve is.
line 394 Ca(OH)2 should be Ca(OH)2
line 426 Ca(OH)2 should be Ca(OH)2
And once again I turn to the XRD part.
Lines 351, 354
It is incorrect to name XRD patterns as XRD spectra. Spectrum is the dependence of intensity from the wavelengh. In XRD we deal with 2theta angle, not the wavelengh. Please, correct.
Figure 10b still doesn't make sense. Just remove it. Figure 10a is quite enough to describe the phase composition. Paragraph 2.3.1 should also be corrected.
Author Response
Please see the attachment.

Reviewer 3 Report (New Reviewer)
Comments and Suggestions for Authors
The manuscript presents valuable experimental results regarding the influence of curing temperature on the mechanical properties and microstructure of low-heat Portland cement concrete. I appreciate the authors' efforts in addressing my previous comments and improving the clarity of the manuscript. However, one major concern remains unresolved.
While the maturity-strength relationship is now thoroughly analyzed, the discussion section remains narrowly focused only on the maturity index, without adequately interpreting or contextualizing the other experimental results obtained in the study.
I strongly recommend extending the discussion to cover these aspects, including:
-
Interpretation of strength development and modulus of elasticity results – for example, how the observed trends compare to existing literature on LHC concretes, and what implications they have for crack resistance or structural design.
-
Analysis of pore structure development (MIP) and its influence on mechanical performance, especially in the later stages – how does the observed pore size reduction and porosity evolution correlate with strength gain?
-
Insights from XRD and SEM analyses – especially regarding the formation of C-S-H gel and reduction of Ca(OH)â‚‚ content at 90 days. These findings are important and should be interpreted with respect to hydration kinetics and durability.
Overall, the manuscript would benefit significantly from a more comprehensive and critical discussion of the experimental results beyond the maturity index. Once this is addressed, I believe the paper will be much stronger and more informative for both researchers and practitioners.
Author Response
Please see the attachment.

This manuscript is a resubmission of an earlier submission. The following is a list of the peer review reports and author responses from that submission.
Round 1
Reviewer 1 Report
Comments and Suggestions for Authors
In the article the compressive strength, splitting tensile strength, axial tensile strength, elastic modulus of concrete were studied. It could be interesting to Materilas readers but unfortunatelly the paper is poorly edited . The level of English also lacks quality.
There are lots of strange signs looking like reverse comma. Please, correct!
Line 12, 42 for example.
Lines 27 - 29. Spaces between some words are missing.
Line 43 reference [1212] has to be [1].
The physical methods description is absolutely incompetent. What devices and settings have the authors used? What software and databases were used?
I suppose that Materials readers are unfamiliar with such abbrevations as C2S. When first mentioned C2S should be writtern as chemical formula Caâ‚‚SiOâ‚„ with mineral name and Powder Diffraction Files database nubmer.
It is incorrect to name XRD patterns as XRD spectra. Spectrum is the dependence of intensity from the wavelengh. In XRD we deal with 2theta angle, not the wavelengh. Please, correct.
The figure 10 on the right differs from the classic XRD patern on the left. Can you explane it? It looks like calculated XRD pattern with enormously big FWHM (peak width) value.
line 314 Should be: Figure 16. Energy
Comments on the Quality of English Language
Capital letters after commas are confusing. Sometimes the meaning of the text escapes altogether.
Reviewer 2 Report
Comments and Suggestions for Authors
After carefully reading and analyzing the manuscript in each of its sections, I have formulated the following observations and suggestions, which, in my opinion, are aimed at improving the clarity, accuracy, and coherence of the document to ensure it meets the required quality standards for publication.
Abstract
- The abstract lacks a clear flow. It jumps between different topics (experimental setup, results, conclusions, and analysis methods) without adequate transitions. This makes it harder to follow the main message of the paper. Consider reorganizing the information for better coherence. Start by briefly stating the purpose of the study, followed by methods, results, and conclusions.
- The phrase "standard maintenance and simulated field actual temperature maintenance" is unclear. What do "maintenance" and "temperature maintenance" refer to? It would be helpful to explain this briefly to provide more context.
- The abstract briefly mentions that maturity indexes were calculated using three formulas (N-S, F-P, and D-L). It would be helpful to add more context on how these formulas are applied or why they were chosen for this study.
- The mention of "fitting logarithmic function, exponential function, and hyperbolic function" is vague. Specify what data were fitted using these functions (e.g., strength vs. maturity, temperature vs. compressive strength), and briefly explain why these functions were selected.
- The statement "the strength of low heat Portland cement concrete increases with the increase of maturity under quasi environmental curing and standard curing" is a good conclusion, but it would be helpful to add a bit more detail about how the curing conditions compare and how significant the results were. For instance, "The strength of low-heat Portland cement concrete increases with maturity, with quasi-environmental curing showing similar trends to standard curing, but at different rates."
- The phrase "the fitting accuracy of hyperbolic function in F-P equivalent age and D-L equivalent age is the highest" could benefit from specifying what "highest accuracy" means in terms of the results or the error margins.
Introduction
- The introduction contains numerous ideas, but they are presented in a disjointed manner, making it difficult to follow the logical progression of the argument. Consider restructuring the paragraph to provide a clearer flow of information. You might start by introducing the significance of water conveyance tunnels, the challenges of crack resistance, and the role of low heat Portland cement in addressing these challenges.
- The introduction references several technical terms, but these are often not explained in a way that a broader audience might understand. For example:
"Low hydration heat" and "temperature rise inside concrete" are mentioned without clarifying their significance in the context of concrete performance.
The term "maturity method" is mentioned without explanation. A brief definition or context for this term would make it more accessible to a wider audience.
- References to Studies: The introduction cites a long list of studies, but many of the results from these studies are not sufficiently summarized or connected to the current study’s objectives. It is essential to briefly highlight the findings of these studies and explain how they relate to the current research.
- The introduction repeats similar ideas, particularly regarding the properties of low heat Portland cement. For instance, the sentences about low early strength and high late strength are presented multiple times in different ways. This could be condensed for clarity and to avoid redundancy.
- The discussion on "strength development" and "early age strength" appears more than once. These ideas should be consolidated into a single coherent argument.
- There are abrupt transitions between different topics in the introduction. For example, after discussing previous studies on the mechanical properties of low heat cement, the introduction jumps to a discussion of the maturity method and then the proposed test without a smooth transition. This can be improved by adding transitional phrases that tie the information together and indicate how each section relates to the broader research goals.
- The purpose of the study is introduced toward the end of the section ("This test will pass the standard maintenance..."), but it could be stated more clearly and earlier on. The introduction should briefly outline the research gap and explain how this study aims to fill that gap, ideally before delving into specific methodologies.
Experiment
- The description of raw materials can benefit from further elaboration. For instance, the characteristics of the low heat Portland cement could be briefly explained, as well as the choice of aggregates (e.g., why natural sand and pebbles were selected).
- The fly ash is mentioned without providing any details on its origin or specific properties. Providing more context about its grade and source would help understand its role in the mix.
- The curing conditions are mentioned briefly, but the temperature conditions, the rationale behind them, and how they relate to real-world conditions could be elaborated. For example: "The specimens were subjected to standard curing at 20°C, as well as quasi-environmental temperature maintenance that simulates site-specific conditions." The term "quasi-environmental temperature" should be defined or explained more clearly.
- The term "quasi-environmental maintenance" is used but not defined. It would be helpful to provide a brief description of how this maintenance was implemented and why it was chosen.
- Microscopic testing (XRD, SEM) is mentioned briefly but is not elaborated on. It would be useful to clarify how these tests contribute to understanding the concrete’s mechanical properties and what specific phenomena they are designed to examine. For example, explain what the XRD and SEM analysis will reveal about the concrete's microstructure.
Experiment Results and Analysis
- The section presents results for various tests (compressive strength, elastic modulus, splitting tensile, axial tensile, etc.) in a linear format, which is acceptable, but a brief introductory paragraph for each subsection would help orient the reader. For example, before starting the analysis of compressive strength, briefly introduce what was being tested and the context for the comparison.
- While the results are described in terms of differences between the two curing methods, there is no mention of statistical analysis (e.g., significance testing) to verify if the differences observed are statistically significant. Including a brief mention of the statistical methods used (e.g., t-test, ANOVA) and the corresponding significance values would strengthen the validity of the results.
- The statement "the early strength growth of DRH is better than the experimental value" is somewhat ambiguous. It would be clearer if the experimental values were more explicitly stated. Was this based on expected design values or previous studies? Clarify this comparison.
- It is stated that the growth rate of DRH-II is higher than DRH-I at 3 and 7 days, but the significance of these differences (e.g., their impact on the concrete’s overall performance) should be explored further. Is this a critical difference in terms of structural performance?
- The interpretation of the splitting tensile strength data suggests that temperature has a significant effect on its development, but it would be beneficial to mention how this property contributes to overall crack resistance. Does the improved tensile strength correlate with a decrease in cracking in practical applications?
- The description of the three maturity formulas (N-S, F-P, and D-L) is good, but it would be more informative if you provided a brief explanation of how these models compare in their application to the current data. Why were these models chosen? Are there specific advantages in using them for low heat cement?
- The fitting of the strength-maturity relationship using exponential, logarithmic, and hyperbolic functions is well-described. However, the correlation coefficients are presented in a somewhat overwhelming manner. It would be more readable if they were summarized in a table for easier comparison. Additionally, it would help to explain the physical meaning of the parameters (e.g., the temporal characteristic parameters in the exponential function).
- The description of XRD analysis is generally good, but it would benefit from a clearer explanation of how the results are related to the mechanical properties observed. For instance, how does the presence of C2S or Ca(OH)2 at different curing ages directly correlate with the observed improvements in strength?
- The microstructural analysis via SEM is mentioned but could be elaborated further. For example, discuss how the microstructural changes observed correlate with the macroscopic properties (compressive strength, tensile strength, etc.). What does the development of C-S-H gel suggest about long-term durability and crack resistance?
Conclusion
- The conclusion contains several incomplete or fragmented sentences, which affect clarity and readability. For example:
- "The temperature decreases and decreases" should be rephrased for clarity, such as "As the temperature decreases, the mechanical properties of concrete also decrease."
- "It shows that the research done by the laboratory has a strong reference significance for the actual situation on the spot" could be clearer. Consider: "The findings from the laboratory study are highly relevant and provide valuable insights for real-world applications."
· The points are somewhat disjointed, and the section could benefit from smoother transitions between ideas. For instance, after discussing the effects of temperature, you could transition into the predictive models, followed by the discussion of the microscopic findings.
· The phrase "increase with the increase of temperature" is repeated twice in the first point, which can be simplified to avoid redundancy. For example, "The compressive strength, splitting tensile strength, axial tensile strength, elastic modulus, and their growth rates increase as the temperature rises" conveys the same idea without redundancy.
· Similarly, "temperature decreases and decreases" could be replaced with a more concise statement like "As the temperature decreases, the mechanical properties also decrease."
· The statement "The temperature change has the greatest influence on the splitting tensile strength of concrete" is somewhat vague. It would be better to clarify what aspect of the temperature change has the greatest influence (e.g., the rate of temperature change or the absolute temperature). Additionally, it would be helpful to quantify "greatest influence" by referencing specific data or trends observed.
- The statement "the difference of concrete performance change is within the acceptable range" is vague. What is meant by "acceptable range"? Is this a statistically significant range, or is it based on design standards? Providing more details would make the conclusion more robust.
- The third point states, "The hyperbolic fitting function based on F-P equivalent age and D-L equivalent age can well predict the development law between maturity and strength." This is a solid conclusion, but it would be beneficial to add more specifics. For example, "The hyperbolic fitting function based on F-P and D-L equivalent age models accurately predicts the relationship between concrete maturity and strength, with high correlation coefficients observed for compressive, tensile, and elastic properties.
- The last point mentions the presence of a "large amount of C-S-H gel" but does not explicitly connect this to the observed mechanical properties. It would be more informative to explain how the development of C-S-H gel directly contributes to the mechanical properties or durability of the concrete. For example, "The increased formation of C-S-H gel in low heat cement enhances the concrete’s compactness and contributes to its improved strength and crack resistance over time."
Round 2
Reviewer 1 Report
Comments and Suggestions for Authors
Unfortunatelly, authors ignored most of the crusial recomendations.
The article still requires thorough editing.
I've found no answer what exact devices, settings and software were used by authors (XRD and SEM method).
There is no answer why the lines at the right figure 10 are so broad. Is it a simulatuion?
Further dialogue is only possible if the authors provide raw XRD data related to fig. 10. (*.brml, *.raw, *.xy of dat files).
Comments on the Quality of English Language
Line 34. Good compactness, Ensure the later strength development of concrete.
Perhaps, it should be:
Good compactness ensures the later strength development of concrete.
Line 19, 35, 61, etc.)There should be no caption after comma.
Line 85. Lack of research on the later stage of concrete.
Were is the verb?
line 332. Inhibitory contraction,Improve the durability of concrete.
Perhaps, it should be:
Inhibitory contraction improves the durability of concrete.
These are just a few mistakes. There are probably many more.
Reviewer 2 Report
Comments and Suggestions for Authors
Dear Authors,
After reviewing the revised version of your manuscript, I can confirm that the reviewers' comments have been adequately addressed. The modifications and improvements made have significantly enhanced the clarity and scientific rigor of the document.
Author Response
Thank you very much, Your reply is very important to me. Thank you again for your suggestions for my paper. I will actively learn in the professional field. Will be more rigorous in writing. Strive to achieve better results